# Pivotal Role of mTOR in Non-Skin Manifestations of Psoriasis

**DOI:** 10.3390/ijms25126778

**Published:** 2024-06-20

**Authors:** Ka Joo, Claudio Karsulovic, Milisa Sore, Lia Hojman

**Affiliations:** 1Facultad de Medicina Clínica Alemana de Santiago, Universidad del Desarrollo, Santiago P.O. Box 7630000, Chile; kajoohu@gmail.com; 2Investigation in Dermatology and Autoimmunity—IDeA Lab, Instituto de Ciencias e Innovación en Medicina, Universidad del Desarrollo, Santiago P.O. Box 7630000, Chile; ckarsulovic@alemana.cl (C.K.); milisa.sore@gmail.com (M.S.); 3Rheumatology Section, Internal Medicine Department, Facultad de Medicina Clínica Alemana de Santiago, Universidad del Desarrollo, Santiago P.O. Box 7630000, Chile; 4Dermatology Section, Surgery Department, Facultad de Medicina Clínica Alemana de Santiago, Universidad del Desarrollo, Santiago P.O. Box 7630000, Chile

**Keywords:** mTOR pathway, psoriasis, cardiovascular disease

## Abstract

Psoriasis is a chronic inflammatory condition affecting 2% of the Western population. It includes diverse manifestations influenced by genetic predisposition, environmental factors, and immune status. The sustained activation of mTOR is a key element in psoriasis pathogenesis, leading to an uncontrolled proliferation of cytokines. Furthermore, mTOR activation has been linked with the transition from psoriasis to non-skin manifestations such as psoriatic arthritis and cardiovascular events. While therapies targeting pro-inflammatory cytokines have shown efficacy, additional pathways may offer therapeutic potential. The PI3K/Akt/mTOR pathway, known for its role in cell growth, proliferation, and metabolism, has emerged as a potential therapeutic target in psoriasis. This review explores the relevance of mTOR in psoriasis pathophysiology, focusing on its involvement in cutaneous and atheromatous plaque proliferation, psoriatic arthritis, and cardiovascular disease. The activation of mTOR promotes keratinocyte and synovial cell proliferation, contributing to plaque formation and joint inflammation. Moreover, mTOR activation may exacerbate the cardiovascular risk by promoting pro-inflammatory cytokine production and dysregulation lipid and glucose metabolism. The inhibition of mTOR has shown promise in preclinical studies, reducing skin inflammation and plaque proliferation. Furthermore, mTOR inhibition may mitigate cardiovascular risk by modulating cholesterol metabolism and attenuating atherosclerosis progression. Understanding the role of mTOR in psoriasis, psoriatic arthritis, and cardiovascular disease provides insight into the potential treatment avenues and sheds light on the complex interplay of the immune and metabolic pathways in these conditions.

## 1. About Psoriasis

Psoriasis is a chronic, inflammatory disease with no cure at present. It is one of the most frequent chronic inflammatory skin diseases, with a prevalence of 2% of the Western population [1], and there is evidence that indicates that the prevalence has increased in the last few years [2]. It presents with different manifestations that depend on the interplay of the environment, genetics, and immune status [3], typically affecting the skin and nails. At the same time, approximately 1.3% to 34.7% of individuals with psoriasis develop chronic inflammatory arthritis [1]. It is characterized by erythematous, scaly plaques that can present on any skin surface, and it is associated with the significant deterioration in the quality of life and life expectancy, mainly due to higher cardiovascular disease risk. Skin plaques are caused by epidermal hyperproliferation with abnormal keratinocyte differentiation, inflammation, and increased angiogenesis [1]. Keratinocyte proliferation is induced by IL-17, IL-22, and TNF-α [4], which in turn, when activated, produce and secrete cytokines and inflammatory mediators, with the arrival of inflammatory cells in the dermis and epidermis [5], the immune response, epidermic hyperproliferation, and neovascularization [6]. This process can be treated through the inhibition of IL-17, IL-22, and TNF-α [7]. However, it is possible that other upregulated molecules could also be inhibited and could serve as therapeutic targets. One of them is the PI3K/Akt/mTOR pathway [6].

Herein, we aim to study the relevance of mTOR in psoriasis, specifically in the proliferation of cutaneous and atheromatous plaques, joint involvement in psoriatic arthritis, and its involvement in cardiovascular disease.

### 1.1. Clinical Manifestations

#### 1.1.1. Skin Manifestations

Psoriasis has different subtypes with their respective clinical manifestations (Figure 1). The most prevalent is psoriasis vulgaris, reaching up to 90% of cases [8]; due to this, usually, when psoriasis is mentioned, it refers to this subtype. Psoriasis vulgaris consists of well-defined plaques of chronic duration, which generate itchiness, are erythematous, and are covered with scales. The plaques can cover large areas of skin but usually affect the trunk, extremities, and scalp [8].

Other subtypes of psoriasis with chronic presentation are pustular psoriasis and inverse psoriasis [8]. Pustular psoriasis is characterized by the presence of multiple pustular lesions that coalesce, either in a localized or generalized manner; notably, there are two localized forms, namely palmoplantar pustular and Hallopeau’s acrodermatitis continua [8]. Both of these variants primarily affect the hands and feet but differ in the specific area where they target. Palmoplantar pustular psoriasis, as its name mentions, is confined to the palms and soles, while Hallopeau’s acrodermatitis is limited to the tips of the fingers and toes, including the nails [8]. On the other hand, inverse psoriasis, also known as flexural psoriasis, manifests as erythematous and erosive plaques and patches in skin folds, such as the axillary, intergluteal, inframammary, and genital folds. Inverse psoriasis, also known as flexural psoriasis, affects the folds of the skin (for example, axillary, intergluteal, inframammary, and genital involvement) and is characterized clinically by slightly erosive erythematous plaques and patches [8].

Among the subtypes of the acute presentation of psoriasis are generalized pustular psoriasis, guttate psoriasis, and erythrodermic psoriasis. Generalized pustular psoriasis has a rapidly progressive profile accompanied by systemic symptoms in addition to generalized erythema and subcorneal pustules. Guttate psoriasis occurs mainly in children and adolescents, but one-third of these patients end up progressing to psoriasis vulgaris in adulthood [8]. It is generally triggered by streptococcal tonsillitis and presents as small erythematous plaques. Finally, erythrodermic psoriasis is a condition that can occur in any subtype of psoriasis and requires immediate treatment as it generates erythema and inflammation on up to 90% of the body [8].

#### 1.1.2. Non-Skin Manifestations

Patients with psoriasis have numerous associated manifestations, among which, in this work, we will address psoriatic arthritis and cardiovascular diseases. Psoriatic arthritis can occur in up to 30% of patients with psoriasis [9], particularly in severe cases or those involving the nails or scalp. This chronic inflammatory condition significantly affects the joints, mainly generating arthritis, spondylitis, enthesitis, and dactylitis, which can cause joint deterioration and consequent disability. It has been seen that patients who present with psoriasis or psoriatic arthritis are at a higher risk of developing extra-musculoskeletal inflammation and other comorbidities, such as metabolic syndrome and cardiovascular diseases [9] with a higher prevalence and mortality due to cardiovascular disease due to various mechanisms that we will explain below. Although there is evidence that these patients develop modifiable cardiovascular risk factors (such as hypertension, diabetes, hyperlipidemia, and nonalcoholic fatty liver), psoriasis is still considered an independent cardiovascular risk factor [10].

## 2. Pathophysiology of the Skin and Non-Skin Manifestations in Psoriasis

The pathogenesis of the disease is not entirely elucidated due to its heterogeneity, both in terms of the tissues and pathways involved, as well as its clinical manifestations (Figure 2). Its pathogenesis begins with a genetic predisposition associated with triggers such as cutaneous changes or biomechanical stress, which, through Damage-Associated Molecular Patterns (DAMPs) and Pathogen-Associated Molecular Patterns (PAMPs), activate Toll-Like Receptors (TLRs) on dendritic cells and macrophages to present their antigens via Major Histocompatibility Complex I (MHC I) to T cells, mainly CD8, stimulating the release of various pro-inflammatory cytokines that activate Th1 and Th17 [9]. This leads, as mentioned above, to the release of various molecules, among which we highlight IL-17, IL-22, and TNF-α, which further release pro-inflammatory cells and attract chondrocytes, synovial fibroblasts, osteoclasts, endothelial cells, and keratinocytes [9].

**Genetic predisposition**. Psoriasis has a vital hereditary component, leading to around 17.7% of prevalence in first-degree relatives [11]. The genetic segment of the major histocompatibility complex (MHC), situated on the short arm of chromosome 6, harbors numerous alleles or haplotypes of human leukocyte antigen (HLA) class I. These variants are linked to heightened susceptibility to psoriasis and are also correlated with various clinical characteristics of the condition. Several genetic regions associate psoriasis with psoriatic arthritis, such as *HLA-C*06:02*; this allele demonstrates a greater strength to psoriasis compared to its association with psoriatic arthritis, and additionally, this allele is linked to an early onset of psoriasis and a prolonged interval between the initiation of skin manifestations and joint involvement [11].

As of now, genome-wide association studies have not identified a confirmed genetic link between psoriasis and cardiovascular disease. However, observational studies have indicated elevated homocysteine levels in psoriasis patients due to DNA demethylation [11]. Research suggests that the polymorphisms in methylenetetrahydrofolate reductase can contribute to DNA methylation, and multiple studies have shown homocysteine to be an independent risk factor for cardiovascular disease (CVD) [11].

**Biomechanical stress.** When a joint is exposed to biomechanical overload, the enthesis suffers mechanical stress (microtrauma) that induces the release of cytokines and growth factors. The mechanostimulation of mesenchymal cells induces CXL1 and CCL2, which recruit monocytes that differentiate into osteoclasts, leading to secondary synovitis. Therefore, biomechanical loading is essential for the transition from systemic autoimmunity to joint inflammation [9]. Additionally, a study conducted by Thorarensen et al. concluded that psoriatic patients exposed to physical trauma (which was stratified into subgroups of joint, bone, nerve, and skin trauma) are at an increased risk of developing psoriatic arthritis compared with the psoriatic patient without trauma [12].

### 2.1. Formation of Psoriatic Plaques

IL-17, IL-22, and TNF-α induce and sustain the key components of psoriasis. They induce the proliferation of keratinocytes and alterations in their differentiation, leading to the formation of psoriatic plaques: epidermal acanthosis, hyperkeratosis, and parakeratosis. These pro-inflammatory cytokines contribute to the sustained activation of mTOR, maintaining the massive proliferation of the basal layer and altered differentiation in the aforementioned suprabasal layers [6]. Consequently, activated keratinocytes produce important pro-inflammatory cytokines, and chemokines attract more inflammatory cells from the vascular system. This forms a vicious circle of excessive immune response perpetuating cutaneous and systemic psoriasis changes [6].

### 2.2. Increased Cardiovascular Risk

The excessive immune response plays a crucial role in the association between psoriasis and cardiovascular disease [1]. This is based on the “psoriatic march” hypothesis proposed by Boehncke [1]. This hypothesis postulates that psoriasis is considered a systemic inflammatory condition due to the increased levels of the various systemic biomarkers that it generates. Consequently, this triggers insulin resistance, as evidenced by the HOMA-IR test, and subsequent endothelial dysfunction. The impaired endothelial function results in heightened vascular stiffness due to diminished nitric oxide production, an insulin-dependent process. This serves as the first step of atherosclerosis. Moreover, it has been demonstrated that insulin resistance disrupts epidermal homeostasis, promoting the proliferation and reducing the differentiation of keratinocytes, thereby contributing to the increased formation of psoriatic plaques [1].

Furthermore, the increase in the surrounding cytokines secondary to the excessive immune response in psoriasis directly induces endothelial damage and contributes to the development of an unstable atherosclerotic plaque, characterized by the enhanced lipid uptake by macrophages in the formation of foam cells [13].

### 2.3. Psoriatic Arthritis

In psoriatic arthritis, these pro-inflammatory cytokines, particularly IL-17, promote the activation of chondrocytes, synovial fibroblasts, and osteoclasts, leading to the increased proliferation of synovial tissue and bone resorption. The thickening of the synovial tissue, coupled with the presence of growth factors, results in hypoxia and consequently triggers angiogenesis. The activation of the inflammatory cascade manifests pathologically as synovitis, enthesitis, erosions, and cartilage degradation [9].

Moreover, Adamopoulos et al. demonstrated that IL-17 directly activates osteoclast precursors and exacerbates RANKL-mediated osteoclast genesis, along with intensifying synovial inflammation. This also results in increased bone resorption in inflammatory arthritis [14].

## 3. Key Evidence of the Pivotal Role of mTOR Non-Skin Manifestations

The mammalian target of rapamycin (mTOR) is a kinase that is part of the PI3-K/Akt/mTORC1 pathway, which regulates many cellular processes like growth, proliferation, angiogenesis, and metabolism [15].

It has been shown that the pathway is upregulated in psoriatic epidermis due to its activation, dependent on growth factors and cytokines like IL-17 and IL-22, known psoriasis activators. In these patients, mTOR activation is followed by the excessive proliferation of keratinocytes and synovial cells [16].

The dysregulation of the PI3-K/Akt/mTOR pathway has been demonstrated to impact both the cutaneous structure and its inflammatory mechanisms. Specifically in psoriasis, mTOR activation is associated with various immune cells (Th1, Th17, IL-1b, IL-6, IL-17, and TNF-alpha) acting on keratinocytes, leading to uncontrolled proliferation [17]. Furthermore, mTOR is activated by diverse cytokines such as IL-1b and IL-23, activating Th17 and secreting IL-17 and IL-22. This implies that a persistent activation of mTOR occurs in psoriasis, promoting the formation of psoriatic plaques [17] (Table 1).

### 3.1. mTOR in the Transition from Psoriasis to Psoriatic Arthritis

mTOR also plays a significant role in the skin-to-joint-involvement transition. Studies have shown a significant rise in IL-17A in the synovial membrane and fluid in rheumatoid arthritis (RA) patients, and the skin lesions of psoriatic patients [18,19]. IL-23, which is involved in IL-17 production, stimulates epidermal hyperplasia, synovial inflammation, and bone destruction in RA patients [20,21]. Even more, it has been shown that IL-17A induces osteoclast RANKL-dependent differentiation and worsens the synovial inflammation and bone loss in inflammatory arthritis [13].

In psoriatic arthritis physiopathology, a poorly controlled proliferation of synovial fibroblasts has been shown, which is associated with mTOR activation. Subsequently, the inhibition of the PI3K/Akt/mTOR pathway may delay abnormal immune cell survival, induce the apoptosis of inflamed, hyperplastic synovial cells, suppress the apoptosis of joint chondrocytes, and promote joint cartilage repair [22].

### 3.2. mTOR in Atheromatous Plaque Formation

The association between mTOR pathway activation and atheroma formation has been the subject of study. Research has suggested that mTOR pathway activation could promote the proliferation of macrophages and smooth muscle cells, which are the primary cells of the foam layer [23]. These macrophages migrate towards the intima and express scavenger receptors and phagocytose oxidized lipoprotein, leading in this way to lipid deposition [24]. Also, the overproduction of pro-inflammatory cytokines, such IL-17, is known for its pro-atherogenic activity. IL-17 is essential in forming vicious cycles between the skin and other affected tissues and endothelial cells. The mTOR activation in epithelial cells and resident macrophages in the skin and other psoriasis-affected tissues stimulate the production of IL-17, which also stimulates the production of IL-12 and IL-23, activating the endothelial and resident immune cells to produce TNF-α. With the higher levels of TNF-α [25], both vascular and skin tissues recruit TH17 T lymphocytes, which produce higher levels of IL-17 [25].

On the other hand, the high levels of IL-17 in vascular tissue activate NFκB-dependent pathways, raising the tissue levels of IL-6 and chemokines [26]. Beyond this specific form of cytokine-driven destabilization and inflammation of the endothelium, mTOR has several ways of producing local damage [27]. The activation of the mTORC1 complex can promote the production of IL-12 and IL-23 in dendritic cells, helping to recruit more activated T lymphocytes [27]. Also, mTORC activation participates in the proliferation of smooth muscle cells and regulates oxidative stress, which is frequently enhanced by high levels of IL-6 and TNF-α [28]. Finally, mTOR has been demonstrated to participate in the polarization of circulating monocytes; our group demonstrated that rheumatoid arthritis patients with cardiovascular disease present higher serum levels of phosphorylated S6R protein, part of the mTOR pathway activation. Also, mTOR is essential in the polarization of monocytes to the inflammatory phenotype when circulating before reaching the target tissue [29].

Furthermore, activating mTOR signaling not only exacerbates endothelial dysfunction and fosters the generation of additional foam cells but also enhances the migration and proliferation of vascular smooth muscle cells in the early stages of atherosclerosis. Additionally, it facilitates the development of vulnerable plaques and the replication of vascular smooth muscle cells in the later stages of atherosclerosis [30].

### 3.3. mTOR Inhibition Reduces Plaque Proliferation

When mTOR inhibitors are applied to intact, healthy skin, there is typically a minimal response since proliferation has already occurred. However, in the presence of unhealthy or disrupted skin, the healing process can be impeded or halted, particularly in severe cases. In a recent study, the mTOR inhibitor, rapamycin, was found to inhibit cell proliferation in culture and alleviate symptoms in the animal models of psoriasis induced by imiquimod. Roy et al. demonstrated that the topical application of rapamycin reduced epidermal differentiation in mice induced with psoriatic dermatitis using imiquimod by reducing the activation of the PI3K-mTOR pathway. Another rapamycin analog that has shown potential success for psoriasis treatment is everolimus, combined with tacrolimus, either as a monotherapy or in severe refractory psoriasis [17].

Aramwit et al. conducted a proteomics study suggesting that sericin fibroin (ScF) reduces keratinocyte proliferation via the mTOR pathway by downregulating the mTOR protein. This corresponds to the modulation of TNF-α, Wnt, and IL-1β levels, ultimately promoting an anti-inflammatory environment through IL-17 downregulation [31].

Sericin and fibroin, silk proteins extracted from *Bombyx mori cocoons*, are natural biopolymers with various medicinal properties, particularly renowned for their wound-healing abilities [31]. Cream-based and polyvinyl alcohol-based formulations of sericin have shown efficacy in the preclinical studies related to psoriasis treatment. The primary anti-psoriatic mechanisms of sericin involve inhibiting epidermal cell overgrowth, reducing epidermal inflammation, and restoring epidermal cell homeostasis through immunomodulatory and antioxidative effects [31].

*Delphinidin* (also *delphinidine*), a natural dietary antioxidant, has demonstrated beneficial properties in various human diseases, including inflammatory skin conditions like psoriasis [17]. Roy et al. showcased *delphinidin’s* antipsoriatic effects and identified its therapeutic mechanism as a dual inhibitor of PI3K/Akt and mTOR, effectively targeting mTORC1 and mTORC2. Interestingly, *delphinidin* binds to the same site on mTORC1 as rapamycin but does not require FKBP12 binding to exert its effects. Moreover, *delphinidin* can inhibit PI3K, leading to decreased IL-22 release and the alleviation of psoriatic symptoms [17].

As previously mentioned, insulin resistance is essential in forming psoriatic plaques through disturbing the epidermal homeostasis [1]. The above means that metformin, a primary anti-diabetic medication, could be used to treat psoriasis because it reduces insulin resistance. Recent research has shown that metformin decreases psoriatic plaque proliferation by inhibiting the mTOR pathway. This results in suppressing keratinocyte proliferation and the expression of pro-inflammatory cytokines such as IL-6 and TNF- α [32].

Several investigations have studied natural extracts as a potential treatment for psoriasis by inhibiting the mTOR pathway. One is fisetin, which showed promising results in its studies in mice for psoriasis and other inflammatory skin diseases. Fisetin promotes keratinocyte differentiation and autophagy via the inhibition of IL-17A and the Akt/mTOR pathway [25].

Baicalin has been found effective in alleviating the common psoriasis symptoms, including epidermal thickening, erythema, and desquamation induced by pro-inflammatory cytokines (IL-17A, IL-22, and IL-23) [17]. Another natural extract that has been the subject of study is *Glycyrrhiza glabra*, a strain of Chinese licorice that has antioxidant, antitumor, and anti-inflammatory properties and relieves psoriatic symptoms by inhibiting mTOR which increases regulatory T cells in the spleen [17]. Finally, we mention matrine, derived from *Sophora flavescens Aiton*, which inactivates the PI3K/Akt/mTOR pathway. As well as those that we have indicated above, hundreds of natural extracts that inhibit the mTOR pathway are being studied and could be potential treatments for psoriasis in the future [17].

It has been reported by Dorjsembe et al. that there is a relation between mTOR inhibition and less proliferation of psoriatic skin plaques [33]. The effect on psoriasis of goat beard weed (*Aruncus dioicus var. kamtschaticus*), known in Oriental medicine for its anti-inflammatory, antioxidant and anticarcinogenic [34,35] properties, was studied. The results showed that goat beard weed is able to significantly suppress cutaneous inflammation and improve the skin’s integrity by inhibiting keratinocyte hyperproliferation and reducing psoriatic marker expression. Moreover, it was shown that goat beard weed was able to suppress the activity of the Akt/mTOR and JAK2/JAK3 pathway in HaCat cells, so its effects on psoriasis could be related to this action [33].

The effect of Taiwanofungus camphoratus (*TC*; *Ganoderma comphoratum*; *Antrodia camphorate*; *Antrodia cinnamomea*) in psoriasis has also been studied [36]. It has been shown that TC is able to exert anti-inflammatory, anticarcinogenic, and antiangiogenic effects by reducing IL-17A-mediated inflammation and cell proliferation through the mTOR/p70S6K and NF-κB pathways [37], which suggests mTOR’s relevance in psoriasis pathogenesis and its potential role as a therapeutic target [36].

### 3.4. mTOR Inhibition Reduces the Cardiovascular Risk

It has been reported by multiple sources that psoriatic patients present with a higher cardiovascular risk with up to an excess of 50% [38]. This association seems to relate to the severity of and time from diagnosis; however, the precise underlying mechanism is unknown [1].

Some studies associate psoriasis with cardiovascular disease due to the direct damage caused by the pro-inflammatory state, as previously mentioned. Other studies suggest that the pathogenesis of cardiovascular disease in psoriasis is based on a combination of the activation of both cutaneous and systemic immune systems with added pre-existent cardiometabolic features that trigger endothelial activation and dysfunction [13].

The mTOR pathway can also be implicated in the excess cardiovascular risk in psoriasis. mTOR activation may produce an excessive pro-inflammatory cytokine production, such as IL-17 [39], which is detrimental to the cardiovascular system. Many studies have shown that IL-17 is pro-atherogenic, prothrombotic, and pro-inflammatory and behaves as a cell attractant, which is associated with plaque instability. Furthermore, it has been found that IL-17 inhibition is associated with atherosclerosis regression. Additionally, mTOR activation may influence lipid and glucose metabolism, which contributes to dyslipidemia, diabetes, and obesity [40], which are common risk factors in psoriatic patients.

Recent research indicates that mTOR inhibition can decrease the cardiovascular risk through various mechanisms. For instance, *Huayu Qutan Recipe (HYQT*), a compound of traditional Chinese herbal medicine, exhibits anti-atherosclerosis effects by inhibiting foam cell formation through the promotion of lipophagy and cholesterol efflux via the mTORC1/TFEB/ABCA1-SCARB1 signal axis [41]. Li et al. investigated how both Huayu Qutan Recipe and rapamycin (both mTOR inhibitors) enhance cholesterol efflux. At the same time, *HY1485* (an mTOR agonist) counteracts the effects of *Huayu Qutan Recipe* by reducing cholesterol efflux. This study concluded that *HYQT* may lower the cardiovascular risk by boosting cholesterol efflux and degrading macrophage-derived foam cell formation through the mTOR/TFEB signaling pathway [41].

The role of microRNA (miR)-125b-1-3p (a short noncoding RNA molecule that regulates gene expression) in cardiovascular disease is well documented. However, its specific function in regulating autophagy and its influence on atherosclerosis in vascular smooth muscle cells are not fully understood. Chen et al. conducted a study to investigate this matter and concluded that miR-125b-1-3p plays a crucial role in enhancing the autophagic processes. It also inhibits the foam cell formation in vascular smooth muscle cells and mitigates the progression of atherosclerosis, partly through the modulation of the RRAGD/mTOR/ULK1 signaling axis. Thus, miR-125b-1-3p is a promising target for preventive and therapeutic strategies for atherosclerosis [42].

## 4. Conclusions

Adaptation modules such as mTOR play crucial roles in the cell adaptation to different microenvironmental changes. As discussed here, in the context of the autoimmune activation observed in psoriasis, the secondary involvement of mTOR may exert a significant influence on the cellular adaptations that promote inflammatory responses in distant tissues such the synovium, bone, or vascular endothelium. The interplay of the mTOR signaling cascade could serve as a pivotal nexus in transducing specific autoinflammatory processes occurring in the skin to propagate metastatic inflammation in articular or vascular structures. Recognizing the pivotal role of these proteins within this framework not only enhances our comprehension of these phenomena but also allows us to develop novel therapeutic approaches and strategies targeting this pathway. By specifically targeting mTOR pathways, we may envisage the prospect of ameliorating or preempting the non-skin manifestations associated with psoriasis, thus addressing a significant unmet clinical need in the management of this complex autoimmune disorder.

## Figures and Tables

**Figure 1 ijms-25-06778-f001:**
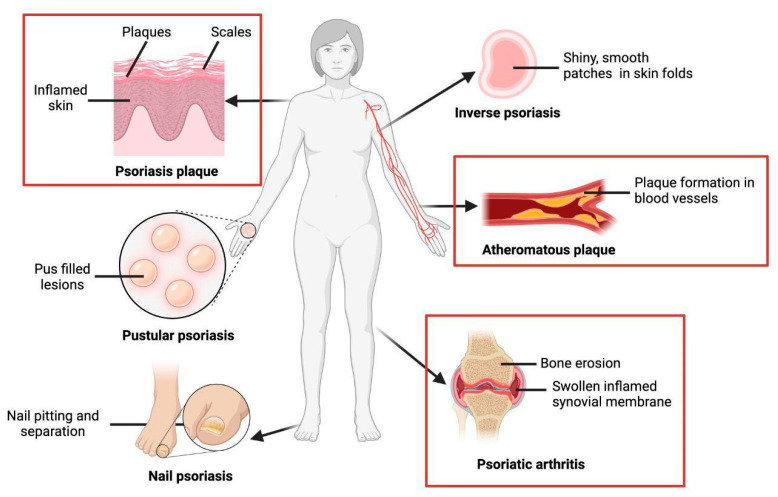
Clinical manifestations of psoriasis. Earmarked in red are the non-skin manifestations associated with mTOR.

**Figure 2 ijms-25-06778-f002:**
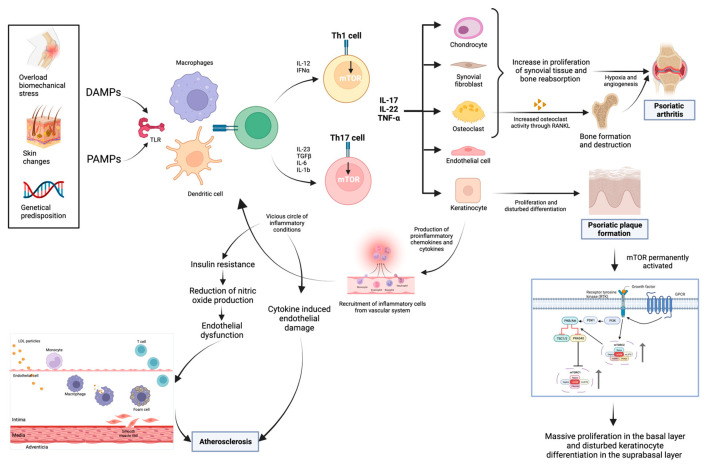
Pathogenesis of psoriasis and its relationship with psoriatic arthritis and cardiovascular disease.

**Table 1 ijms-25-06778-t001:** Implications of the pivotal role of mTOR in non-skin manifestations.

Aspect	Implications
mTOR in the transition from psoriasis to psoriatic arthritis	Contributes to synovial inflammation and bone destruction by inducing IL-17A production and activating the abnormal proliferation of synovial fibroblasts.
mTOR in atheromatous plaque formation	Associated with the proliferation of macrophages and smooth muscle cells, contributing to atheroma formation and the production of pro-inflammatory cytokines.
mTOR inhibition reduces plaque proliferation	The inhibition of mTOR by certain substances has been shown to reduce skin inflammation and keratinocyte hyperproliferation.
mTOR inhibition reduces the cardiovascular risk	mTOR activation may increase the cardiovascular risk by promoting the production of pro-inflammatory cytokines and affecting lipid and glucose metabolism, suggesting a potential role in the pathogenesis of cardiovascular disease in psoriasis patients.

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
