# Peer review of "Pivotal Role of mTOR in Non-Skin Manifestations of Psoriasis"

_ijms, 2024, doi:10.3390/ijms25126778_

Round 1

Reviewer 1 Report

Comments and Suggestions for Authors

This article provides a comprehensive overview of psoriasis and its association with mTOR, as well as the role of mTOR in non-skin manifestations of psoriasis. My suggestion is that the abstract should be improved, the current abstract seems more like an introduction in terms of content and format. The author should succinctly summarize the main content of this review in the abstract. Besides, the first half of the article is excessively lengthy and unrelated to the mTOR signaling. The content related to mTOR signaling only begins in the third section. The author should properly coordinate the structure of the article.

Comments on the Quality of English Language

Minor editing of the English language is required.

Reviewer 2 Report

Comments and Suggestions for Authors

One of my main objections to this article is that it writes about evidence without referring to the appropriate literary sources. In each sentence containing literature data, the source must be indicated.

Just a few examples:

  „At  the same time, approximately 1.3% to 34.7% of individuals with Psoriasis develop chronic inflammatory arthritis.” (line 38-40)

„Guttate psoriasis occurs mainly in children and adolescents, but one-third of these patients end up progressing to psoriasis vulgaris in adulthood.” (line 78-80)

„Psoriatic arthritis can occur in up to 30% of patients with psoriasis, particularly in severe cases or those involving nails or scalp.” (line 86-88)

 „Psoriasis has a vital hereditary component, leading to around 17.7% of prevalence in first-degree relatives.” (line 113-114)

„However, observational studies have indicated elevated homocysteine levels in psoriasis patients due to DNA demethylation.” (line 124-125)

„Specifically in psoriasis, mTOR activation is associated with various immune cells (Th1, Th17, IL-1b, IL-6, IL-17, TNF-alpha) acting on keratinocytes, leading to uncontrolled proliferation.” (line 187-189)

etc.

Spelling is another important thing that should be carefully reviewed in the article.

„Psoriasis” is often written with capital P instead of lowercase p (it also happens int he case of other words that they are written with capital letters instead of lowercase letters).

„MTOR” is written instead of mTOR (line 194)- mTOR is correct even if it is the first word in the sentence

„y” (line 45)- I suppose it’s an „and”

„Genome-wide Association Studies” → „genome-wide association studies”

etc.

Two paragraphs about baicalin (lines 279-283 and lines 284-292) contain a lot of similar information. It would improve the quality of the article if these two paragraphs were made into one.

Round 2

Reviewer 2 Report

Comments and Suggestions for Authors

We got a much higher quality version of the original article. I think this is due to hard work, congratulations!